# ON FEATURE DIVERSITY IN ENERGY-BASED MODELS

## ABSTRACT

Energy-based learning is a powerful learning paradigm that encapsulates various discriminative and generative approaches. An energy-based model (EBM) is typically formed of inner-model(s) that learn a combination of the different features to generate an energy mapping for each input configuration. In this paper, we focus on the diversity of the produced feature set. We extend the probably approximately correct (PAC) theory of EBMs and analyze the effect of redundancy reduction on the performance of EBMs. We derive generalization bounds for various learning contexts, i.e., regression, classification, and implicit regression, with different energy functions and we show that indeed reducing redundancy of the feature set can consistently decrease the gap between the true and empirical expectation of the energy and boosts the performance of the model.

## 1 INTRODUCTION

The energy-based learning paradigm was first proposed by Zhu & Mumford (1998); LeCun et al. (2006) as an alternative to probabilistic graphical models (Koller & Friedman, 2009). As their name suggests, energy-based models (EBMs) map each input 'configuration' to a single scalar, called the 'energy'. In the learning phase, the parameters of the model are optimized by associating the desired configurations with small energy values and the undesired ones with higher energy values (Kumar et al., 2019; Song & Ermon, 2019; Yang et al., 2016). In the inference phase, given an incomplete input configuration, the energy surface is explored to find the remaining variables which yield the lowest energy. EBMs encapsulate solutions to several supervised approaches (LeCun et al., 2006; Fang & Liu, 2016) and unsupervised learning problems (Deng et al., 2020; Bakhtin et al., 2021; Zhao et al., 2020; Xu et al., 2022) and provide a common theoretical framework for many learning models, including traditional discriminative (Zhai et al., 2016; Li et al., 2020) and generative (Zhu & Mumford, 1998; Xie et al., 2017b; Zhao et al., 2017; Che et al., 2020; Khalifa et al., 2021) approaches.

Formally, let us denote the energy function by $E(h, \boldsymbol{x}, \boldsymbol{y})$, where $h = G_{\boldsymbol{W}}(\boldsymbol{x})$ represents the model with parameters $\boldsymbol{W}$ to be optimized during training and $\boldsymbol{x}, \boldsymbol{y}$ are sets of variables. Figure 1 illustrates how classification, regression, and implicit regression can be expressed as EBMs. In Figure 1 (a), a regression scenario is presented. The input $\boldsymbol{x}$, e.g., an image, is transformed using an inner model $G_{\boldsymbol{W}}(\boldsymbol{x})$ and its distance, to the second input $\boldsymbol{y}$ is computed yielding the energy function. A valid energy function in this case can be the $L_1$ or the $L_2$ distance. In the binary classification case (Figure 1 (b)), the energy can be defined as $E(h, \boldsymbol{x}, \boldsymbol{y}) = -y G_{\boldsymbol{W}}(\boldsymbol{x})$. In the implicit regression case (Figure 1 (c)), we have two inner models and the energy can be defined as the $L_2$ distance between their outputs $E(h, \boldsymbol{x}, \boldsymbol{y}) = \frac{1}{2}||G_{\boldsymbol{W}}^{(1)}(\boldsymbol{x}) - G_{\boldsymbol{W}}^{(2)}(\boldsymbol{y})||_2^2$. In the inference phase, given an input $\boldsymbol{x}$, the label $\boldsymbol{y}^*$ can be obtained by solving the following optimization problem:

$$\boldsymbol{y}^* = \arg\min_{\boldsymbol{y}} E(h, \boldsymbol{x}, \boldsymbol{y}). \tag{1}$$

An EBM typically relies on an inner model, i.e., $G_{\boldsymbol{W}}(\boldsymbol{x})$, to generate the desired energy landscape (LeCun et al., 2006). Depending on the problem at hand, this function can be constructed as a linear projection, a kernel method, or a neural network and its parameters are optimized in a data-driven manner in the training phase. Formally, $G_{\boldsymbol{W}}(\boldsymbol{x})$ can be written as

$$G_{\boldsymbol{W}}(\boldsymbol{x}) = \sum_{i}^{D} w_i \phi_i(\boldsymbol{x}), \tag{2}$$

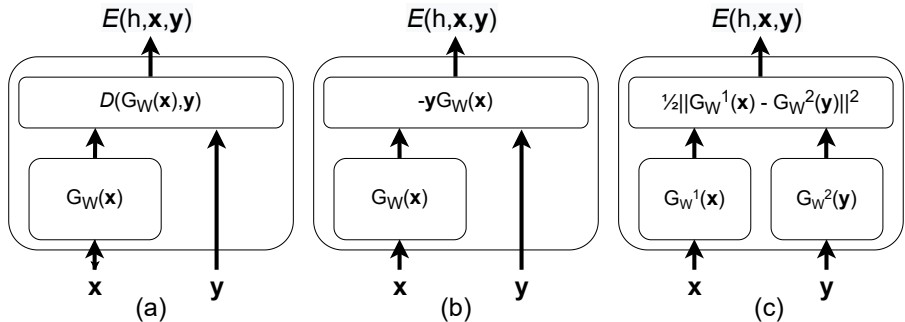

Figure 1: An illustration of energy-based models used to solve (a) a regression problem (b) a binary classification problem (c) an implicit regression problem.

where $\{\phi_1(\cdot), \cdots, \phi_D(\cdot)\}$ is the feature set, which can be hand-crafted, separately trained from unlabeled data (Zhang & LeCun, 2017), or modeled by a neural network and optimized in the training phase of the EBM model (Xie et al., 2016; Yu et al., 2020; Xie et al., 2021). In the rest of the paper, we assume that the inner models $G_{\boldsymbol{W}}$ defined in the energy-based learning system (Figure 1) are obtained as a weighted sum of different features as expressed in equation 2.

In (Zhang, 2013), it was shown that simply minimizing the empirical energy over the training data does not theoretically guarantee the minimization of the expected value of the true energy. Thus, developing and motivating novel regularization techniques is required (Zhang & LeCun, 2017). We argue that the quality of the feature set $\{\phi_1(\cdot), \cdots, \phi_D(\cdot)\}$ plays a critical role in the overall performance of the global model. In this work, we extend the theoretical analysis of (Zhang, 2013) and focus on the 'diversity' of this set and its effect on the generalization ability of the EBM models. Intuitively, it is clear that a less correlated set of intermediate representations is richer and thus able to capture more complex patterns in the input. Thus, it is important to avoid redundant features for achieving a better performance. However, a theoretical analysis is missing. We start by quantifying the diversity of a set of feature functions. To this end, we introduce $\vartheta - \tau$-diversity:

**Definition 1** (($\vartheta - \tau$)-diversity). *A set of feature functions, $\{\phi_1(\cdot), \cdots, \phi_D(\cdot)\}$ is called $\vartheta$-diverse, if there exists a constant $\vartheta \in \mathbb{R}$, such that for every input $\boldsymbol{x}$ we have*

$$\frac{1}{2} \sum_{i \neq j}^{D} (\phi_i(\boldsymbol{x}) - \phi_j(\boldsymbol{x}))^2 \geq \vartheta^2 \tag{3}$$

*with a high probability $\tau$.*

Intuitively, if two feature maps $\phi_i(\cdot)$ and $\phi_j(\cdot)$ are non-redundant, they have different outputs for the same input with a high probability. However, if, for example, the features are extracted using a neural network with a ReLU activation function, there is a high probability that some of the features associated with the input will be zero. Thus, defining a lower bound for the pair-wise diversity directly is impractical. Therefore, we quantify diversity as the lower-bound over the sum of the pair-wise distances of the feature maps as expressed in equation 3 and $\vartheta$ measures the diversity of a set.

In machine learning context, diversity has been explored in ensemble learning (Li et al., 2012; Yu et al., 2011; Li et al., 2017), sampling (Derezinski et al., 2019; Bıyık et al., 2019), ranking (Wu et al., 2019; Qin & Zhu, 2013), pruning (Singh et al., 2020; Lee et al., 2020), and neural networks (Xie et al., 2015; Shen et al., 2021). In Xie et al. (2015; 2017a), it was shown theoretically and experimentally that avoiding redundancy over the weights of a neural network using the mutual angles as a diversity measure improves the generalization ability of the model. In this work, we explore a new line of research, where diversity is defined over the feature maps directly, using the $(\vartheta - \tau)$-diversity, in the context of energy-based learning. In (Zhao et al., 2017), a similar idea was empirically explored. A "repelling regularizer" was proposed to force non-redundant or orthogonal feature representations. Moreover, the idea of learning while avoiding redundancy has been used recently in the context of semi-supervised learning (Zbontar et al., 2021; Bardes et al., 2021). Reducing redundancy by minimizing the cross-correlation of features learned using a Siamese network

(Zbontar et al., 2021) was empirically shown to improve the generalization ability, yet a theoretical analysis to prove this has so far been lacking.

In this paper, we close the gap between empirical experience and theory. We theoretically study the generalization ability of EBMs in different learning contexts, i.e., regression, classification, implicit regression, and we derive new generalization bounds using the $(\vartheta-\tau)$-diversity providing theoretical guarantees that avoiding redundancy indeed improves the generalization ability of the model. The contributions of this paper can be summarized as follows:

- We explore a new line of research, where diversity is defined over the features representing the input data and not over the model's parameters. To this end, we introduce $(\vartheta - \tau)$-diversity as a quantification of the diversity of a given feature set.
- We extend the theoretical analysis (Zhang, 2013) and study the effect of avoiding redundancy of a feature set on the generalization of EBMs (Lemmas 3 to 7 and Theorem 1 to 5).
- We derive bounds for the expectation of the true energy in different learning contexts, i.e., regression, classification, and implicit regression, using different energy functions. Our analysis consistently shows that avoiding redundancy by increasing the diversity of the feature set can boost the performance of an EBM.

## 2 PAC-LEARNING OF EBMS WITH $(\vartheta - \tau)$-DIVERSITY

In this section, we derive a qualitative justification for $(\vartheta-\tau)$-diversity using probably approximately correct (PAC) learning (Valiant, 1984; Mohri et al., 2018; Li et al., 2019). The PAC-based theory for standard EBMs has been established in (Zhang, 2013). First, we start by defining Rademacher complexity:

**Definition 2.** *(Bartlett & Mendelson, 2002; Mohri et al., 2018) For a given dataset with $m$ samples $\boldsymbol{S} = \{\boldsymbol{x}_i, y_i\}_{i=1}^m$ from a distribution $\mathcal{D}$ and for a model space $\mathcal{F} : \mathcal{X} \to \mathbb{R}$ with a single dimensional output, the Empirical Rademacher complexity $\hat{\mathcal{R}}_m(\mathcal{F})$ of the set $\mathcal{F}$ is defined as follows:*

$$\hat{\mathcal{R}}_m(\mathcal{F}) = \mathbb{E}_\sigma \left[ \sup_{f \in \mathcal{F}} \frac{1}{m} \sum_{i=1}^m \sigma_i f(\boldsymbol{x}_i) \right], \tag{4}$$

*where the Rademacher variables $\sigma = \{\sigma_1, \cdots, \sigma_m\}$ are independent uniform random variables in $\{-1, 1\}$.*

The Rademacher complexity $\mathcal{R}_m(\mathcal{F})$ is defined as the expectation of the Empirical Rademacher complexity over training set, i.e., $\mathcal{R}_m(\mathcal{F}) = \mathbb{E}_{\boldsymbol{S} \sim \mathcal{D}^m}[\hat{\mathcal{R}}_m(\mathcal{F})]$. Based on this quantity, (Bartlett & Mendelson, 2002), several learning guarantees for EBMs have been shown (Zhang, 2013). We recall the following two lemmas related to the estimation error and the Rademacher complexity. In Lemma 2, we present the principal PAC-learning bound for energy functions with finite outputs.

**Lemma 1.** *(Wolf, 2018) For $\mathcal{F} \in \mathbb{R}^\mathcal{X}$, assume that $g : \mathbb{R} \to \mathbb{R}$ is a $L_g$-Lipschitz continuous function and $\mathcal{A} = \{g \circ f : f \in \mathcal{F}\}$. We have*

$$\mathcal{R}_m(\mathcal{A}) \leq L_g \mathcal{R}_m(\mathcal{F}). \tag{5}$$

**Lemma 2.** *(Zhang, 2013) For a well-defined energy function $E(h, \boldsymbol{x}, \boldsymbol{y})$ over hypothesis class $\mathcal{H}$, input set $\mathcal{X}$ and output set $\mathcal{Y}$ (LeCun et al., 2006), the following holds for all $h$ in $\mathcal{H}$ with a probability of at least $1 - \delta$*

$$\mathbb{E}_{(\boldsymbol{x},\boldsymbol{y}) \sim \boldsymbol{D}}[E(h, \boldsymbol{x}, \boldsymbol{y})] \leq \frac{1}{m} \sum_{(\boldsymbol{x},\boldsymbol{y}) \in \boldsymbol{S}} E(h, \boldsymbol{x}, \boldsymbol{y}) + 2\mathcal{R}_m(\mathcal{E})$$

$$+ M\sqrt{\frac{\log(2/\delta)}{2m}}, \tag{6}$$

*where $\mathcal{E}$ is the energy function class defined as $\mathcal{E} = \{E(h, \boldsymbol{x}, \boldsymbol{y})|h \in \mathcal{H}\}$, $\mathcal{R}_m(\mathcal{E})$ is its Rademacher complexity, and $M$ is the upper bound of $\mathcal{E}$.*

Lemma 2 provides a generalization bound for EBMs with well-defined (non-negative) and bounded energy. The expected energy is bounded using the sum of three terms: The first term is the empirical expectation of energy over the training data, the second term depends on the Rademacher complexity of the energy class, and the third term involves the number of the training data $m$ and the upper-bound of the energy function $M$. This shows that merely minimizing the empirical expectation of energy, i.e., the first term, may not yield a good approximation of the true expectation. In (Zhang & LeCun, 2017), it has been shown that regularization using unlabeled data reduces the second and third terms leading to better generalization. In this work, we express these two terms using the $(\vartheta - \tau)$-diversity and show that employing a diversity strategy may also decrease the gap between the true and empirical expectation of the energy. In Section 2.1, we consider the special case of regression and derive two bounds for two energy functions based on $L_1$ and $L_2$ distances. In Section 2.2, we derive a bound for the binary classification task using as energy function $E(h, \boldsymbol{x}, \boldsymbol{y}) = -yG_{\boldsymbol{W}}(\boldsymbol{x})$ (LeCun et al., 2006). In Section 2.3, we consider the case of implicit regression, which encapsulates different learning problems such as metric learning, generative models, and denoising (LeCun et al., 2006). For this case, we use the $L_2$ distance between the inner models as the energy function. In the rest of the paper, we denote the generalization gap, $\mathbb{E}_{(\boldsymbol{x},\boldsymbol{y})\sim\boldsymbol{D}}[E(h, \boldsymbol{x}, \boldsymbol{y})] - \frac{1}{m}\sum_{(\boldsymbol{x},\boldsymbol{y})\in\boldsymbol{S}} E(h, \boldsymbol{x}, \boldsymbol{y})$ by $\Delta_{\boldsymbol{D},\boldsymbol{S}}E$. All the proofs are presented in the supplementary material.

## 2.1 REGRESSION TASK

Regression can be formulated as an energy-based learning problem (Figure 1 (a)) using the inner model $h(\boldsymbol{x}) = G_{\boldsymbol{W}}(\boldsymbol{x}) = \sum_{i=1}^{D} w_i\phi_i(\boldsymbol{x}) = \boldsymbol{w}^T\Phi(\boldsymbol{x})$. We assume that the feature set is positive and well-defined over the input domain $\mathcal{X}$, i.e., $\forall \boldsymbol{x} \in \mathcal{X} : ||\Phi(\boldsymbol{x})||_2 \leq A$, the hypothesis class can be defined as follows: $\mathcal{H} = \{h(\boldsymbol{x}) = G_{\boldsymbol{W}}(\boldsymbol{x}) = \sum_{i=1}^{D} w_i\phi_i(\boldsymbol{x}) = \boldsymbol{w}^T\Phi(\boldsymbol{x}) \mid \Phi \in \mathcal{F}, \forall \boldsymbol{x} : ||\Phi(\boldsymbol{x})||_2 \leq A\}$, the output set $\mathcal{Y} \subset \mathbb{R}$ is bounded, i.e., $y < B$, and the feature set $\{\phi_1(\cdot), \cdots, \phi_D(\cdot)\}$ is $\vartheta$-diverse with a probability $\tau$. The two valid energy functions which can be used for regression are $E_2(h, \boldsymbol{x}, \boldsymbol{y}) = \frac{1}{2}||G_{\boldsymbol{W}}(\boldsymbol{x}) - y||_2^2$ and $E_1(h, \boldsymbol{x}, \boldsymbol{y}) = ||G_{\boldsymbol{W}}(\boldsymbol{x}) - y||_1$ (LeCun et al., 2006). We study these two cases separately and we show theoretically that for both energy functions avoiding redundancy improves generalization of the EBM model.

ENERGY FUNCTION: $E_2$

In this subsection, we present our theoretical analysis on the effect of diversity on the generalization ability of an EBM defined with the energy function $E_2(h, \boldsymbol{x}, \boldsymbol{y}) = \frac{1}{2}||G_{\boldsymbol{W}}(\boldsymbol{x}) - y||_2^2$. We start by the following two Lemmas 3 and 4.

**Lemma 3.** *With a probability of at least $\tau$, we have*

$$\sup_{\boldsymbol{x}, \boldsymbol{W}} |h(\boldsymbol{x})| \leq ||\boldsymbol{w}||_\infty \sqrt{(DA^2 - \vartheta^2)}. \tag{7}$$

**Lemma 4.** *With a probability of at least $\tau$, we have*

$$\sup_{\boldsymbol{x}, y, h} |E(h, \boldsymbol{x}, \boldsymbol{y})| \leq \frac{1}{2}(||\boldsymbol{w}||_\infty \sqrt{(DA^2 - \vartheta^2)} + B)^2. \tag{8}$$

*Proof.* We have $\sup_{\boldsymbol{x}, y, h} |h(\boldsymbol{x}) - y| \leq \sup_{\boldsymbol{x}, y, h}(|h(\boldsymbol{x})| + |y|) = (||\boldsymbol{w}||_\infty \sqrt{DA^2 - \vartheta^2} + B)$. Thus $sup_{x,y,h}|E(h, x, y)| \leq \frac{1}{2}(||\boldsymbol{w}||_\infty \sqrt{DA^2 - \vartheta^2} + B)^2$. $\qquad \square$

Lemmas 3 and 4 bound the supremum of the output of the inner model and the energy function as a function of $\vartheta$, respectively. As it can been seen, both terms are decreasing with respect to diversity. Next, we bound the Rademacher complexity of the energy class, i.e., $\mathcal{R}_m(\mathcal{E})$.

**Lemma 5.** *With a probability of at least $\tau$, we have*

$$\mathcal{R}_m(\mathcal{E}) \leq 2D||\boldsymbol{w}||_\infty(||\boldsymbol{w}||_\infty \sqrt{(DA^2 - \vartheta^2)} + B)\mathcal{R}_m(\mathcal{F}). \tag{9}$$

Lemma 5 expresses the bound of the Rademacher complexity of the energy class using the diversity constant and the Rademacher complexity of the features. Having expressed the different terms of Lemma 2 using diversity, we now present our main result for an energy-basel model trained defined using $E_2$. The main result is presented in Theorem 1.

**Theorem 1.** *For the energy function $E(h, \boldsymbol{x}, \boldsymbol{y}) = \frac{1}{2}||G_{\boldsymbol{W}}(\boldsymbol{x}) - y||_2^2$, over the input set $\mathcal{X} \in \mathbb{R}^N$, hypothesis class $\mathcal{H} = \{h(\boldsymbol{x}) = G_{\boldsymbol{W}}(\boldsymbol{x}) = \sum_{i=1}^D w_i \phi_i(\boldsymbol{x}) = \boldsymbol{w}^T \Phi(\boldsymbol{x}) \mid \Phi \in \mathcal{F}, \forall \boldsymbol{x} : ||\Phi(\boldsymbol{x})||_2 \leq A\}$, and output set $\mathcal{Y} \subset \mathbb{R}$, if the feature set $\{\phi_1(\cdot), \cdots, \phi_D(\cdot)\}$ is $\vartheta$-diverse with a probability $\tau$, with a probability of at least $(1 - \delta)\tau$, the following holds for all $h$ in $\mathcal{H}$:*

$$\Delta_{\boldsymbol{D}, \boldsymbol{S}} E \leq 4D ||\boldsymbol{w}||_\infty (||\boldsymbol{w}||_\infty \sqrt{DA^2 - \vartheta^2} + B) \mathcal{R}_m(\mathcal{F})$$

$$+ \frac{1}{2} (||\boldsymbol{w}||_\infty \sqrt{DA^2 - \vartheta^2} + B)^2 \sqrt{\frac{\log(2/\delta)}{2m}}, \tag{10}$$

*where $B$ is the upper-bound of $\mathcal{Y}$, i.e., $y \leq B, \forall y \in \mathcal{Y}$.*

Theorem 1 express the special case of Lemma 2 using the $(\vartheta - \tau)$-diversity of the feature set $\{\phi_1(\cdot), \cdots, \phi_D(\cdot)\}$. As it can been seen, the bound of the generalization error is inversely proportional to $\vartheta^2$. This theoretically shows that reducing redundancy, i.e., increasing $\vartheta$, reduces the gap between the true and the empirical energies and improves the generalization performance of the EBMs.

ENERGY FUNCTION: $E_1$

In this subsection, we consider the second case of regression using the energy function $E_1(h, \boldsymbol{x}, \boldsymbol{y}) = ||G_{\boldsymbol{W}}(\boldsymbol{x}) - y||_1$. Similar to the previous case, we start by deriving bounds for the energy function and the Rademacher complexity of the class using diversity in Lemmas 6 and 7.

**Lemma 6.** *With a probability of at least $\tau$, we have*

$$\sup_{\boldsymbol{x}, y, h} |E(h, \boldsymbol{x}, \boldsymbol{y})| \leq (||\boldsymbol{w}||_\infty \sqrt{DA^2 - \vartheta^2} + B). \tag{11}$$

**Lemma 7.** *With a probability of at least $\tau$, we have*

$$\mathcal{R}_m(\mathcal{E}) \leq 2D ||\boldsymbol{w}||_\infty \mathcal{R}_m(\mathcal{F}). \tag{12}$$

Next, we derive the main result of the generalization of the EBMs defined using the energy function $E_1$. The main finding is presented in Theorem 2.

**Theorem 2.** *For the energy function $E(h, \boldsymbol{x}, \boldsymbol{y}) = ||G_{\boldsymbol{W}}(\boldsymbol{x}) - y||_1$, over the input set $\mathcal{X} \in \mathbb{R}^N$, hypothesis class $\mathcal{H} = \{h(\boldsymbol{x}) = G_{\boldsymbol{W}}(\boldsymbol{x}) = \sum_{i=1}^D w_i \phi_i(\boldsymbol{x}) = \boldsymbol{w}^T \Phi(\boldsymbol{x}) \mid \Phi \in \mathcal{F}, \forall \boldsymbol{x} ||\Phi(\boldsymbol{x})||_2 \leq A\}$, and output set $\mathcal{Y} \subset \mathbb{R}$, if the feature set $\{\phi_1(\cdot), \cdots, \phi_D(\cdot)\}$ is $\vartheta$-diverse with a probability $\tau$, then with a probability of at least $(1 - \delta)\tau$, the following holds for all $h$ in $\mathcal{H}$:*

$$\Delta_{\boldsymbol{D}, \boldsymbol{S}} E \leq 4D ||\boldsymbol{w}||_\infty \mathcal{R}_m(\mathcal{F}) + (||\boldsymbol{w}||_\infty \sqrt{DA^2 - \vartheta^2} + B) \sqrt{\frac{\log(2/\delta)}{2m}}, \tag{13}$$

*where $B$ is the upper-bound of $\mathcal{Y}$, i.e., $y \leq B, \forall y \in \mathcal{Y}$.*

Similar to Theorem 1, in Theorem 2, we consistently find that the bound of the true expectation of the energy is a decreasing function with respect to $\vartheta$. This proves that for the regression task reducing redundancy can improve the generalization performance of the energy-based model.

## 2.2 BINARY CLASSIFIER

Here, we consider the problem of binary classification, as illustrated in Figure 1 (b). Using the same assumption as in regression for the inner model, i.e., $h(\boldsymbol{x}) = G_{\boldsymbol{W}}(\boldsymbol{x}) = \sum_{i=1}^D w_i \phi_i(\boldsymbol{x}) = \boldsymbol{w}^T \Phi(\boldsymbol{x})$, energy function of $E(h, \boldsymbol{x}, \boldsymbol{y}) = -\mathrm{y} G_{\boldsymbol{W}}(\boldsymbol{x})$ (LeCun et al., 2006), and the $(\vartheta - \tau)$-diversity of the feature set, we express Lemma 2 for this specific configuration in Theorem 3.

**Theorem 3.** *For the energy function $E(h, \boldsymbol{x}, \boldsymbol{y}) = -\mathrm{y} G_{\boldsymbol{W}}(\boldsymbol{x})$, over the input set $\mathcal{X} \in \mathbb{R}^N$, hypothesis class $\mathcal{H} = \{h(\boldsymbol{x}) = G_{\boldsymbol{W}}(\boldsymbol{x}) = \sum_{i=1}^D w_i \phi_i(\boldsymbol{x}) = \boldsymbol{w}^T \Phi(\boldsymbol{x}) \mid \Phi \in \mathcal{F}, \forall \boldsymbol{x} : ||\Phi(\boldsymbol{x})||_2 \leq A\}$, and output set $\mathcal{Y} \subset \mathbb{R}$, if the feature set $\{\phi_1(\cdot), \cdots, \phi_D(\cdot)\}$ is $\vartheta$-diverse with a probability $\tau$, then with a probability of at least $(1 - \delta)\tau$, the following holds for all $h$ in $\mathcal{H}$:*

$$\Delta_{\boldsymbol{D}, \boldsymbol{S}} E \leq 4D ||\boldsymbol{w}||_\infty \mathcal{R}_m(\mathcal{F}) + ||\boldsymbol{w}||_\infty \sqrt{DA^2 - \vartheta^2} \sqrt{\frac{\log(2/\delta)}{2m}}. \tag{14}$$

Similar to the regression task, we note that the upper-bound of the true expectation is a decreasing function with respect to the diversity term. Thus, a less redundant feature set, i.e., higher $\vartheta$, has a lower upper-bound for the true energy.

## 2.3 IMPLICIT REGRESSION

In this section, we consider the problem of implicit regression. This is a general formulation of a different set of problems such as metric learning, where the goal is to learn a distance function between two domains, image denoising, object detection as illustrated in (LeCun et al., 2006), or semi-supervised learning (Zbontar et al., 2021). This form of EBM (Figure 1 (c)) has two inner models, $G_W^1(\cdot)$ and $G_W^2(\cdot)$, which can be equal or different according to the problem at hand. Here, we consider the general case, where the two models correspond to two different combinations of different features, i.e., $G_W^{(1)}(\boldsymbol{x}) = \sum_{i=1}^{D^{(1)}} w_i^{(1)} \phi_i^{(1)}(\boldsymbol{x})$ and $G_W^{(2)}(\boldsymbol{y}) = \sum_{i=1}^{D^{(2)}} w_i^{(2)} \phi_i^{(2)}(\boldsymbol{y})$. Thus, we have a different $(\vartheta - \tau)$-diversity term for each set. The final result is presented in Theorem 4.

**Theorem 4.** *For the energy function $E(h, \boldsymbol{x}, \boldsymbol{y}) = \frac{1}{2}||G_W^{(1)}(\boldsymbol{x}) - G_W^{(2)}(\boldsymbol{y})||_2^2$, over the input set $\mathcal{X} \in \mathbb{R}^N$, hypothesis class $\mathcal{H} = \{h^{(1)}(\boldsymbol{x}) = G_W^{(1)}(\boldsymbol{x}) = \sum_{i=1}^{D^{(1)}} w_i^{(1)} \phi_i^{(1)}(\boldsymbol{x}) = \boldsymbol{w}^{(1)^T} \Phi^{(1)}(\boldsymbol{x}), h^{(2)}(\boldsymbol{x}) = G_W^{(2)}(\boldsymbol{y}) = \sum_{i=1}^{D^{(2)}} w_i^{(2)} \phi_i^{(2)}(\boldsymbol{y}) = \boldsymbol{w}^{(2)^T} \Phi^{(2)}(\boldsymbol{y}) \mid \Phi^{(1)} \in \mathcal{F}_1, \Phi^{(2)} \in \mathcal{F}_2, \forall \boldsymbol{x} : ||\Phi^{(1)}(\boldsymbol{x})||_2 \leq A^{(1)}, \forall \boldsymbol{y} : ||\Phi^{(2)}(\boldsymbol{y})||_2 \leq A^{(2)}\}$, and output set $\mathcal{Y} \subset \mathbb{R}^N$, if the feature set $\{\phi_1^{(1)}(\cdot), \cdots, \phi_{D^{(1)}}^{(1)}(\cdot)\}$ is $\vartheta^{(1)}$-diverse with a probability $\tau_1$ and the feature set $\{\phi_1^{(2)}(\cdot), \cdots, \phi_{D^{(2)}}^{(2)}(\cdot)\}$ is $\vartheta^{(2)}$-diverse with a probability $\tau_2$, then with a probability of at least $(1 - \delta)\tau_1\tau_2$, the following holds for all $h$ in $\mathcal{H}$:*

$$\Delta_{\boldsymbol{D},\boldsymbol{S}} E \leq 8(\sqrt{\mathcal{J}_1} + \sqrt{\mathcal{J}_2})\Big(D^{(1)}||\boldsymbol{w}^{(1)}||_\infty \mathcal{R}_m(\mathcal{F}_1) + D^{(2)}||\boldsymbol{w}^{(2)}||_\infty \mathcal{R}_m(\mathcal{F}_2)\Big)$$

$$+ (\mathcal{J}_1 + \mathcal{J}_2)\sqrt{\frac{\log(2/\delta)}{2m}}, \qquad (15)$$

*where $\mathcal{J}_1 = ||\boldsymbol{w}^{(1)}||_\infty^2 \big(D^{(1)} A^{(1)^2} - \vartheta^{(1)^2}\big)$ and $\mathcal{J}_2 = ||\boldsymbol{w}^{(2)}||_\infty^2 \big(D^{(2)} A^{(2)^2} - \vartheta^{(2)^2}\big)$.*

The upper-bound of the energy model depends on the diversity variable of both feature sets. Moreover, we note that the bound for the implicit regression decreases proportionally to $\vartheta^2$, as opposed to the classification case for example, where the bound is proportional to $\vartheta$. Thus, we can conclude that reducing redundancy improves the generalization of EBM in the implicit regression context.

## 2.4 GENERAL DISCUSSION

We note that the theory developed in our paper (Theorems 1 to 4) is agnostic to the loss function (LeCun et al., 2006) or the optimization strategy used (Kumar et al., 2019; Song & Ermon, 2019; Yu et al., 2020; Xu et al., 2022). We show that reducing the redundancy of the features consistently decreases the upper-bound of the true expectation of the energy and, thus, can boost the generalization performance of the energy-based model. It also should be noted that $A$, i.e., the upper bound of the features and $\vartheta$ are connected. But our findings can be interpreted as follows: given two models with the same value of $A$ (maximum $L_2$ norm of the features), the model with higher diversity $\vartheta$ has a lower generalization bound and is likely to generalize better. We note that our analysis is independent of how the features are obtained, e.g., handcrafted or optimized. In fact, in the recent state-of-the-art EBMs (Khalifa et al., 2021; Bakhtin et al., 2021; Yu et al., 2020), the features are typically parameterized using a deep learning model and optimized during training. Our contribution is twofold. First, we provide theoretical guarantees that reducing redundancy in the feature space can indeed improve the generalization of the EBM. This can pave the way toward providing theoretical guarantees for WORKS ON SELF-SUPERVISED LEARNING using redundancy reduction Zbontar et al. (2021); Bardes et al. (2021); Zhao et al. (2017). Second, our theory can be used to motivate novel redundancy reduction strategies, for example, in the form of regularization, to avoid learning redundant features. Such strategies can improve the performance of the model and improve generalization.

## 3 SIMPLE REGULARIZATION ALGORITHM

In general, theoretical generalization bounds can be too loose to be direct practical implications (Zhang et al., 2017; Neyshabur et al., 2017). However, they typically suggest a regularizer to promote some desired aspects of the hypothesis class (Xie et al., 2015; Li et al., 2019; Kawaguchi et al., 2017). Accordingly, inspired by the theoretical analysis in Section 2, we propose a straightforward



Figure 2: From left to right: (a): 2-D swiss roll ground truth distribution, (b) Distribution learned using a standard EBM model, (b) Distribution learned with augmented loss using our regularizer. Relative to the ground truth, the Jensen-Shannon distance of the standard EBM distribution and ours are 0.27426 and 0.2733, respectively.

strategy to avoid learning redundant features by regularizing the model during the training using a term inversely proportional to $\vartheta - \tau$-diversity of the features. Given an EBM model with a learnable feature set $\{\phi_1(\cdot), \cdots, \phi_D(\cdot)\}$ and a training set $S$, we propose to augment the original training loss $L$ as follows:

$$L_{aug} = L - \beta \sum_{\boldsymbol{x} \in S} \sum_{i \neq j}^{D} (\phi_i(\boldsymbol{x}) - \phi_j(\boldsymbol{x}))^2, \tag{16}$$

where $\beta$ is a hyper-parameter controlling the contribution of the second term in the total loss. The additional term penalizes the similarities between the distinct features ensuring learning a diverse and non-redundant mapping of the data. As a result, this can improve the general performance of our model.

## 3.1 TOY EXAMPLE

We test our regularization strategy first using a toy data. We use an EBM model to learn the distribution of a 2-D Swiss roll illustrated in Figure 2 (a). For the EBM, we use a fully connected neural network composed of two intermediate layers with 1000 units and ReLu activations. We train the models using Stochastic Gradient Langevin Dynamics (SGLD) sampling and the contrastive divergence-like algorithm proposed in (Du & Mordatch, 2019). The total objective of the standard EBM is expressed as follows:

$$L = \frac{1}{N} \sum_n \left( \alpha \big( E(\boldsymbol{x}_n^+)^2 + E(\boldsymbol{x}_n^-)^2 \big) + E(\boldsymbol{x}_n^+) - E(\boldsymbol{x}_n^-) \right), \tag{17}$$

where $\boldsymbol{x}_n^+$ denote positive samples and $\boldsymbol{x}_n^-$ negative samples. We augment this loss using equation 16, i.e., the features are the latent representations obtained at the last intermediate layer.

The distribution learned using both the standard and the proposed approach are illustrated using the kernel density estimation (Terrell & Scott, 1992) in Figure 2. As it can be seen, avoiding redundancy boosts the performance of the EBM model. Indeed, by comparing the two learned distributions, the EBM trained with our approach led to a better approximation of the ground-truth distribution and was able to better capture the tail of the distribution as opposed to the original EBM.

## 3.2 IMAGE GENERATION EXAMPLE

Recently, there has been a high interest in using EBMs to solve image/text generation tasks Du & Mordatch (2019); Du et al. (2021); Khalifa et al. (2021); Deng et al. (2020). In this subsection, we validate the proposed regularizer on the simple example of MNIST digits image generation, as in (Du & Mordatch, 2019). For the EBM model, we use a simple CNN model composed of four convolutional layers followed by a linear layer. The training protocol is the same as in (UvA; Du & Mordatch, 2019), i.e., using Langevin dynamics Markov chain Monte Carlo (MCMC) and a sampling buffer to accelerate training. The full details are available in the supplementary material. In this example, the features, i.e., the latent representation obtained at the last intermediate layer, are learned in an end-to-end way. We evaluate the performance of our approach by augmenting the contrastive divergence loss using equation 16 to penalize the feature redundancy.

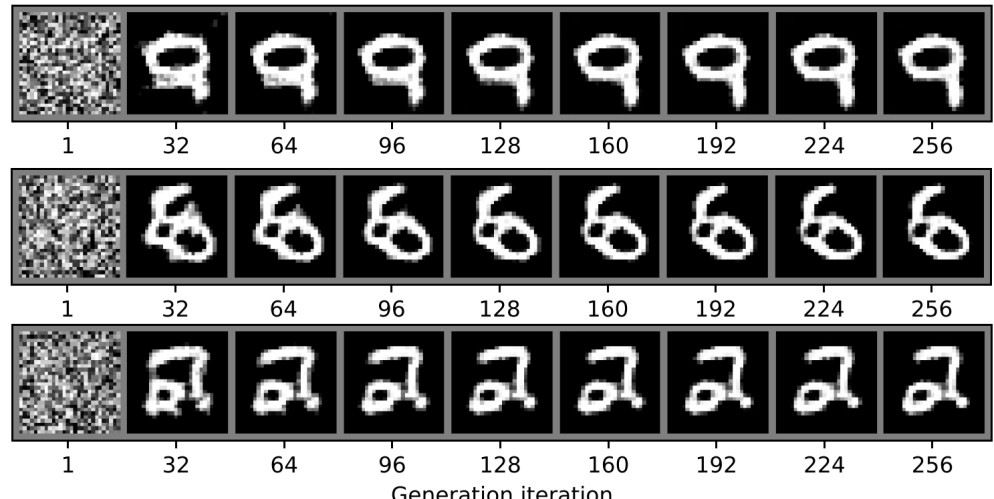

Figure 3: Qualitative results of our approach ($\beta = 1e^{-13}$) : Few intermediate samples of the MCMC sampling (Langevin Dynamics).

| Approach | FID | NLL loss |
|---|---|---|
| EBM | $0.01085 \pm 0.00037$ | $0.71124 \pm 0.01901$ |
| ours ($\beta = 1e^{-11}$) | $0.01071 \pm 0.00040$ | $0.71089 \pm 0.01106$ |
| ours ($\beta = 1e^{-12}$) | $0.01040 \pm 0.00034$ | $\mathbf{0.71052 \pm 0.01118}$ |
| ours ($\beta = 1e^{-13}$) | $\mathbf{0.00985 \pm 0.00058}$ | $0.71076 \pm 0.01105$ |

Table 1: Table of FID scores and negative log-likelihood (NLL) loss of different approaches for generations of MNIST images. Each experiment was performed three times with different random seeds, the results are reported as the mean/SEM over these runs.

We quantitatively evaluate image quality of EBMs with 'Fréchet Inception Distance' (FID) score (Heusel et al., 2017) and the negative log-likelihood (NLL) loss in Table 1 for different values of $\beta$. We note that we obtain consistently better FID and NLL scores by penalizing the similarity of the learned features. The best performance is achieved by $\beta = 1e^{-13}$, which yields more than 10%, in terms of FID, improvement compared to the original EBM model. To gain insights into the visual performance of our approach, we plot a few intermediate samples of the MCMC sampling (Langevin Dynamics). The results obtained by the EBM with $\beta = 1e^{-13}$ are presented in Figure 3. Initiating from random noise, MCMC obtains reasonable figures after only 64 steps. The digits get clearer and more realistic over the iterations. More results are presented in the supplementary material.

## 3.3 CONTINUAL LEARNING EXAMPLE

In this subsection, we validate the proposed regularizer on the Continual Learning (CL) problem. CL tackles the problem of catastrophic forgetting in deep learning models (Parisi et al., 2019; Li & Hoiem, 2017; Shibata et al., 2021). Its main goal is to solve several tasks sequentially without forgetting knowledge learned from the past. So, a continual learner is expected to learn a new task, crucially, without forgetting previous tasks. Recently, an EBM-based CL approach was proposed in (Li et al., 2020) and led to superior results compared to standard approaches. We use the same models and the same experimental protocol used in (Li et al., 2020). However, here we focus only on the class-incremental learning task using CIFAR10 and CIFAR100. We evaluate the performance of our proposed regularizer using both the boundary-aware and boundary-agnostic settings. As defined in (Li et al., 2020), the boundary-aware refers to the situation where the sequence of the tasks has explicit separation between them which is known to the model. The boundary agnostic case refers to the situation where the data distributions gradually changes without a notion of task boundaries.

| Method | Boundary-aware | | Boundary-agnostic | |
| | CIFAR10 | CIFAR100 | CIFAR10 | CIFAR100 |
|---|---|---|---|---|
| EBM | $39.15 \pm 0.86$ | $29.02 \pm 0.24$ | $48.40 \pm 0.80$ | $34.78 \pm 0.26$ |
| ours ($\beta = 1e^{-11}$) | $39.61 \pm 0.81$ | $29.15 \pm 0.27$ | $49.63 \pm 0.90$ | $34.86 \pm 0.30$ |
| ours ($\beta = 1e^{-12}$) | $\mathbf{40.64 \pm 0.79}$ | $\mathbf{29.38 \pm 0.21}$ | $\mathbf{50.25 \pm 0.63}$ | $\mathbf{35.20 \pm 0.23}$ |
| ours ($\beta = 1e^{-13}$) | $40.15 \pm 0.87$ | $29.28 \pm 0.28$ | $50.20 \pm 0.94$ | $35.03 \pm 0.21$ |

Table 2: Evaluation of class-incremental learning on both the boundary-aware and boundary-agnostic setting on CIFAR10 and CIFAR100 datasets. Each experiment was performed ten times with different random seeds, the results are reported as the mean/SEM over these runs.

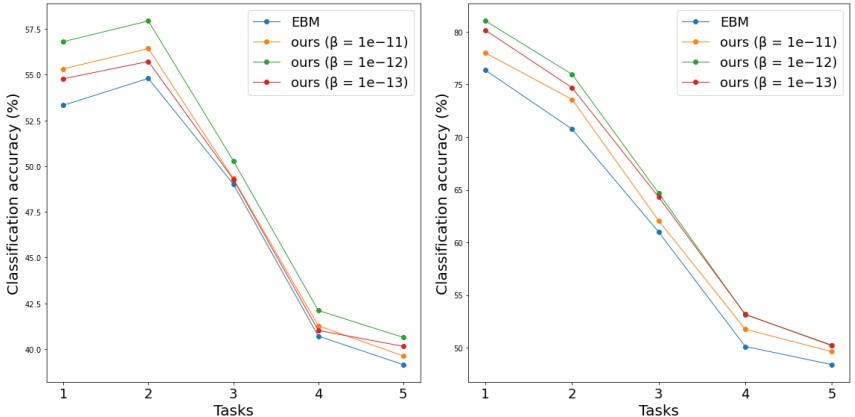

Figure 4: Average test classification accuracy vs number of observed tasks on CIFAR10 using the boundary-aware (left) and boundary-agnostic (right) setting. The results are averaged over ten random seeds.

Similar to Section 3.2, we consider as 'features' the representation obtained by the last intermediate layer. The proposed regularizer is applied on top of this representation. In Table 2, we report the performance of the EBM trained using the original loss and using the loss augmented with our additional term for different values of $\beta$. As shown in Table 2, penalizing feature similarity and promoting the diversity of the feature set boosts the performance of the EBM model and consistently leads to a superior accuracy for both datasets. In Figure 4, we display the accumulated classification accuracy, averaged over tasks, on the test set. Along the five tasks, our approach maintains higher classification accuracy than the standard EBM for both the boundary-aware and boundary-agnostic settings.

## 4 CONCLUSION

Energy-based learning is a powerful learning paradigm that encapsulates various discriminative and generative systems. An EBM is typically formed of one (or many) inner models which learn a combination of different features to generate an energy mapping for each input configuration. In this paper, we introduced a feature diversity concept, i.e., $(\vartheta - \tau)$-diversity, and we used it to extend the PAC theory of EBMs. We derived different generalization bounds for various learning contexts, i.e., regression, classification, and implicit regression, with different energy functions and we consistently found that reducing the redundancy of the feature set can improve the generalization error of energy-based approaches. We also note that our theory is independent of the loss function or the training strategy used to optimize the parameters of the EBM. This provides theoretical guarantees on learning via feature redundancy reduction. Our preliminary experimental results confirm that this is indeed a promising research direction and can motivate developing other approaches to promoting the diversity of the feature set. Future direction include more extensive experimental evaluation of different feature redundancy reduction approaches.

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

## APPENDIX

## A    PROOF OF LEMMA 3

**Lemma** With a probability of at least $\tau$, we have

$$\sup_{\boldsymbol{x}, \boldsymbol{W}} |h(\boldsymbol{x})| \leq ||\boldsymbol{w}||_\infty \sqrt{(DA^2 - \vartheta^2)}, \tag{18}$$

where $A = \sup_{\boldsymbol{x}} ||\phi(\boldsymbol{x})||_2$.

*Proof.*

$$h^2(\boldsymbol{x}) = \left( \sum_{i=1}^{D} w_i \phi_i(\boldsymbol{x}) \right)^2 \leq \left( \sum_{i=1}^{D} ||\boldsymbol{w}||_\infty \phi_i(\boldsymbol{x}) \right)^2 = ||\boldsymbol{w}||_\infty^2 \left( \sum_{i=1}^{D} \phi_i(\boldsymbol{x}) \right)^2$$

$$= ||\boldsymbol{w}||_\infty^2 \left( \sum_{i,j} \phi_i(\boldsymbol{x}) \phi_j(\boldsymbol{x}) \right) = ||\boldsymbol{w}||_\infty^2 \left( \sum_i \phi_i(\boldsymbol{x})^2 + \sum_{i \neq j} \phi_i(\boldsymbol{x}) \phi_j(\boldsymbol{x}) \right) \tag{19}$$

We have $||\Phi(\boldsymbol{x})||_2 \leq A$. For the first term in equation 19, we have $\sum_m \phi_m(\boldsymbol{x})^2 \leq A^2$. By using the identity $\phi_m(\boldsymbol{x}) \phi_n(\boldsymbol{x}) = \frac{1}{2} \left( \phi_m(\boldsymbol{x})^2 + \phi_n(\boldsymbol{x})^2 - (\phi_m(\boldsymbol{x}) - \phi_n(\boldsymbol{x}))^2 \right)$, the second term can be rewritten as

$$\sum_{m \neq n} \phi_m(\boldsymbol{x}) \phi_n(\boldsymbol{x}) = \frac{1}{2} \sum_{m \neq n} \left( \phi_m(\boldsymbol{x})^2 + \phi_n(\boldsymbol{x})^2 - \left( \phi_m(\boldsymbol{x}) - \phi_n(\boldsymbol{x}) \right)^2 \right). \tag{20}$$

In addition, we have with a probability $\tau$, $\frac{1}{2}\sum_{m \neq n}(\phi_m(\boldsymbol{x}) - \phi_n(\boldsymbol{x}))^2 \geq \vartheta^2$. Thus, we have with a probability at least $\tau$:

$$\sum_{m \neq n} \phi_m(\boldsymbol{x})\phi_n(\boldsymbol{x}) \leq \frac{1}{2}(2(D-1)A^2 - 2\vartheta^2) = (D-1)A^2 - \vartheta^2. \tag{21}$$

By putting everything back to equation 19, we have with a probability $\tau$,

$$G_{\boldsymbol{W}}^2(\boldsymbol{x}) \leq ||\boldsymbol{w}||_\infty^2 \Big( A^2 + (D-1)A^2 - \vartheta^2 \Big) = ||\boldsymbol{w}||_\infty^2 (DA^2 - \vartheta^2). \tag{22}$$

Thus, with a probability $\tau$,

$$\sup_{\boldsymbol{x},\boldsymbol{W}} |h(\boldsymbol{x})| \leq \sqrt{\sup_{\boldsymbol{x},\boldsymbol{W}} G_{\boldsymbol{W}}^2(\boldsymbol{x})} \leq ||\boldsymbol{w}||_\infty \sqrt{DA^2 - \vartheta^2}. \tag{23}$$

$\square$

# B    PROOF OF LEMMA 4

**Lemma** With a probability of at least $\tau$, we have

$$\sup_{\boldsymbol{x},y,h} |E(h,\boldsymbol{x},\boldsymbol{y})| \leq \frac{1}{2}(||\boldsymbol{w}||_\infty \sqrt{(DA^2 - \vartheta^2)} + B)^2. \tag{24}$$

*Proof.* We have $\sup_{\boldsymbol{x},y,h} |h(\boldsymbol{x}) - y| \leq \sup_{\boldsymbol{x},y,h}(|h(\boldsymbol{x})| + |y|) = (||\boldsymbol{w}||_\infty \sqrt{DA^2 - \vartheta^2} + B)$. Thus $sup_{x,y,h}|E(h,\boldsymbol{x},\boldsymbol{y})| \leq \frac{1}{2}(||\boldsymbol{w}||_\infty\sqrt{DA^2 - \vartheta^2} + B)^2$. $\square$

# C    PROOF OF LEMMA 5

**Lemma** With a probability of at least $\tau$, we have

$$\mathcal{R}_m(\mathcal{E}) \leq 2D||\boldsymbol{w}||_\infty(||\boldsymbol{w}||_\infty \sqrt{(DA^2 - \vartheta^2)} + B)\mathcal{R}_m(\mathcal{F}) \tag{25}$$

*Proof.* Using the decomposition property of the Rademacher complexity (if $\phi$ is a $L$-Lipschitz function, then $\mathcal{R}_m(\phi(\mathcal{A})) \leq L\mathcal{R}_m(\mathcal{A})$) and given that $\frac{1}{2}||.-y||^2$ is $K$-Lipschitz with a constant $K = sup_{\boldsymbol{x},y,h}||h(\boldsymbol{x}) - y|| \leq (||\boldsymbol{w}||_\infty\sqrt{DA^2 - \vartheta^2} + B)$, we have $\mathcal{R}_m(\mathcal{E}) \leq K\mathcal{R}_m(\mathcal{H}) = (||\boldsymbol{w}||_\infty\sqrt{DA^2 - \vartheta^2} + B)\mathcal{R}_m(\mathcal{H})$, where $\mathcal{H} = \{G_{\boldsymbol{W}}(\boldsymbol{x}) = \sum_{i=1}^D w_i\phi_i(\boldsymbol{x})\}$. We also know that $||\boldsymbol{w}||_1 \leq D||\boldsymbol{w}||_\infty$. Next, similar to the proof of Theorem 2.10 in (Wolf, 2018), we note that $\sum_{i=1}^D w_i\phi_i(\boldsymbol{x}) \in (D||\boldsymbol{w}||_\infty)conv(\mathcal{F} + -(\mathcal{F})) := \mathcal{G}$, where $conv$ denotes the convex hull and $\mathcal{F}$ is the set of $\phi$ functions. Thus, $\mathcal{R}_m(\mathcal{H}) \leq \mathcal{R}_m(\mathcal{G}) = D||\boldsymbol{w}||_\infty\mathcal{R}_m(conv(\mathcal{F} + (-\mathcal{F})) = D||\boldsymbol{w}||_\infty\mathcal{R}_m(\mathcal{F} + (-\mathcal{F})) = 2D||\boldsymbol{w}||_\infty\mathcal{R}_m(\mathcal{F})$. $\square$

# D    PROOF OF THEOREM 1

**Theorem** For the energy function $E(h,\boldsymbol{x},\boldsymbol{y}) = \frac{1}{2}||G_{\boldsymbol{W}}(\boldsymbol{x}) - y||_2^2$, over the input set $\mathcal{X} \in \mathbb{R}^N$, hypothesis class $\mathcal{H} = \{h(\boldsymbol{x}) = G_{\boldsymbol{W}}(\boldsymbol{x}) = \sum_{i=1}^D w_i\phi_i(\boldsymbol{x}) = \boldsymbol{w}^T\Phi(\boldsymbol{x}) \mid \Phi \in \mathcal{F}, \forall \boldsymbol{x} : ||\Phi(\boldsymbol{x})||_2 \leq A\}$, and output set $\mathcal{Y} \subset \mathbb{R}$, if the feature set $\{\phi_1(\cdot), \cdots, \phi_D(\cdot)\}$ is $\vartheta$-diverse with a probability $\tau$, with a probability of at least $(1 - \delta)\tau$, the following holds for all h in $\mathcal{H}$:

$$\mathbb{E}_{(\boldsymbol{x},\boldsymbol{y}) \sim \boldsymbol{D}}[E(h,\boldsymbol{x},\boldsymbol{y})] \leq \frac{1}{m}\sum_{(\boldsymbol{x},\boldsymbol{y}) \in \boldsymbol{S}} E(h,\boldsymbol{x},\boldsymbol{y}) + 4D||\boldsymbol{w}||_\infty(||\boldsymbol{w}||_\infty\sqrt{DA^2 - \vartheta^2} + B)\mathcal{R}_m(\mathcal{F})$$

$$+ \frac{1}{2}(||\boldsymbol{w}||_\infty\sqrt{DA^2 - \vartheta^2} + B)^2\sqrt{\frac{\log(2/\delta)}{2m}}, \quad (26)$$

where B is the upper-bound of $\mathcal{Y}$, i.e., $y \leq B, \forall y \in \mathcal{Y}$.

*Proof.* We replace the variables in Lemma 1 using Lemma 4 and Lemma 5. $\square$

## E    PROOF OF LEMMA 6

**Lemma** With a probability of at least $\tau$, we have

$$\sup_{\boldsymbol{x},y,h} |E(h,\boldsymbol{x},\boldsymbol{y})| \leq (||\boldsymbol{w}||_\infty \sqrt{DA^2 - \vartheta^2} + B). \tag{27}$$

*Proof.* We have $\sup_{\boldsymbol{x},y,h} |h(\boldsymbol{x}) - y| \leq \sup_{\boldsymbol{x},y,h}(|h(\boldsymbol{x})| + |y|) = (||\boldsymbol{w}||_\infty \sqrt{DA^2 - \vartheta^2} + B).$    □

## F    PROOF OF LEMMA 7

**Lemma** With a probability of at least $\tau$, we have

$$\mathcal{R}_m(\mathcal{E}) \leq 2D||\boldsymbol{w}||_\infty \mathcal{R}_m(\mathcal{F}) \tag{28}$$

*Proof.* $|.|$ is 1-Lipschitz, Thus $\mathcal{R}_m(\mathcal{E}) \leq \mathcal{R}_m(\mathcal{H})$.    □

## G    PROOF OF THEOREM 2

**Theorem** For the energy function $E(h,\boldsymbol{x},\boldsymbol{y}) = ||G_{\boldsymbol{W}}(\boldsymbol{x}) - y||_1$, over the input set $\mathcal{X} \in \mathbb{R}^N$, hypothesis class $\mathcal{H} = \{h(\boldsymbol{x}) = G_{\boldsymbol{W}}(\boldsymbol{x}) = \sum_{i=1}^D w_i \phi_i(\boldsymbol{x}) = \boldsymbol{w}^T \Phi(\boldsymbol{x}) \mid \Phi \in \mathcal{F}, \forall \boldsymbol{x} \, ||\Phi(\boldsymbol{x})||_2 \leq A\}$, and output set $\mathcal{Y} \subset \mathbb{R}$, if the feature set $\{\phi_1(\cdot), \cdots, \phi_D(\cdot)\}$ is $\vartheta$-diverse with a probability $\tau$, then with a probability of at least $(1-\delta)\tau$, the following holds for all h in $\mathcal{H}$:

$$\mathbb{E}_{(\boldsymbol{x},\boldsymbol{y})\sim\boldsymbol{D}}[E(h,\boldsymbol{x},\boldsymbol{y})] \leq \frac{1}{m}\sum_{(\boldsymbol{x},\boldsymbol{y})\in\boldsymbol{S}} E(h,\boldsymbol{x},\boldsymbol{y}) + 4D||\boldsymbol{w}||_\infty \mathcal{R}_m(\mathcal{F})$$

$$+ (||\boldsymbol{w}||_\infty \sqrt{DA^2 - \vartheta^2} + B)\sqrt{\frac{\log(2/\delta)}{2m}}, \tag{29}$$

where B is the upper-bound of $\mathcal{Y}$, i.e., $y \leq B, \forall y \in \mathcal{Y}$.

*Proof.* We replace the variables in Lemma 1 using Lemma 6 and Lemma 7.    □

## H    PROOF OF THEOREM 3

**Lemma 8.** *With a probability of at least $\tau$, we have*

$$\sup_{\boldsymbol{x},y,h} |E(h,\boldsymbol{x},\boldsymbol{y})| \leq ||\boldsymbol{w}||_\infty \sqrt{DA^2 - \vartheta^2}. \tag{30}$$

*Proof.* We have $\sup -y G_{\boldsymbol{W}}(\boldsymbol{x}) \leq \sup |G_{\boldsymbol{W}}(\boldsymbol{x})| \leq ||\boldsymbol{w}||_\infty \sqrt{DA^2 - \vartheta^2}.$    □

**Lemma 9.** *With a probability of at least $\tau$, we have*

$$\mathcal{R}_m(\mathcal{E}) \leq 2D||\boldsymbol{w}||_\infty \mathcal{R}_m(\mathcal{F}) \tag{31}$$

*Proof.* We note that for $y \in \{-1, 1\}$, $\sigma$ and $-y\sigma$ follow the same distribution. Thus, we have $\mathcal{R}_m(\mathcal{E}) = \mathcal{R}_m(\mathcal{H})$. Next, we note that $\mathcal{R}_m(\mathcal{H}) \leq 2D||\boldsymbol{w}||_\infty \mathcal{R}_m(\mathcal{F})$.    □

**Theorem 3** For a well-defined energy function $E(h, \boldsymbol{x}, \boldsymbol{y})$ (LeCun et al., 2006), over hypothesis class $\mathcal{H}$, input set $\mathcal{X}$ and output set $\mathcal{Y}$, if it has upper-bound M, then with a probability of at least $1 - \delta$, the following holds for all h in $\mathcal{H}$

$$\mathbb{E}_{(\boldsymbol{x},\boldsymbol{y})\sim\boldsymbol{D}}[E(h,\boldsymbol{x},\boldsymbol{y})] \leq \frac{1}{m}\sum_{(\boldsymbol{x},\boldsymbol{y})\in\boldsymbol{S}} E(h,\boldsymbol{x},\boldsymbol{y}) + 4D||\boldsymbol{w}||_\infty \mathcal{R}_m(\mathcal{F})$$

$$+ ||\boldsymbol{w}||_\infty \sqrt{DA^2 - \vartheta^2}\sqrt{\frac{\log(2/\delta)}{2m}}, \tag{32}$$

*Proof.* We replace the variables in Lemma 1 using Lemma 8 and Lemma 9.    □

## I   Proof of Theorem 4

**Lemma 10.** *With a probability of at least $\tau_1\tau_2$, we have*

$$\sup_{\boldsymbol{x},y,h} |E(h,\boldsymbol{x},\boldsymbol{y})| \leq \left(\mathcal{J}_1 + \mathcal{J}_2\right) \tag{33}$$

*Proof.* We have $||G_{\boldsymbol{W}}^{(1)}(\boldsymbol{x}) - G_{\boldsymbol{W}}^{(2)}(\boldsymbol{y})||_2^2 \leq 2(||G_{\boldsymbol{W}}^{(1)}(\boldsymbol{x})||_2^2 + ||G_{\boldsymbol{W}}^{(2)}(\boldsymbol{y})||_2^2)$. Similar to Theorem 1, we have $\sup ||G_{\boldsymbol{W}}^{(1)}(\boldsymbol{x})||_2^2 \leq ||\boldsymbol{w}^{(1)}||_\infty^2 \left(D^{(1)}A^{(1)^2} - \vartheta^{(1)^2}\right) = \mathcal{J}_1$ and $\sup ||G_{\boldsymbol{W}}^{(2)}(\boldsymbol{y})||_2^2 \leq ||\boldsymbol{w}^{(2)}||_\infty^2 \left(D^{(2)}A^{(2)^2} - \vartheta^{(2)^2}\right) = \mathcal{J}_2$. We also have $E(h,\boldsymbol{x},\boldsymbol{y}) = \frac{1}{2}||G_{\boldsymbol{W}}^{(1)}(\boldsymbol{x}) - G_{\boldsymbol{W}}^{(2)}(\boldsymbol{y})||_2^2$.   $\square$

**Lemma 11.** *With a probability of at least $\tau_1\tau_2$, we have*

$$\mathcal{R}_m(\mathcal{E}) \leq 4(\sqrt{\mathcal{J}_1} + \sqrt{\mathcal{J}_2})\left(D^{(1)}||\boldsymbol{w}^{(1)}||_\infty\mathcal{R}_m(\mathcal{F}_1) + D^{(2)}||\boldsymbol{w}^{(2)}||_\infty\mathcal{R}_m(\mathcal{F}_2)\right) \tag{34}$$

*Proof.* Let $f$ be the square function, i.e., $f(x) = \frac{1}{2}x^2$ and $\mathcal{E}_0 = \{G_{\boldsymbol{W}}^{(1)}(x) - G_{\boldsymbol{W}}^{(2)}(y) \mid x \in \mathcal{X}, y \in \mathcal{Y}\}$. We have $\mathcal{E} = f(\mathcal{E}_0 + (-\mathcal{E}_0))$. $f$ is Lipschitz over the input space, with a constant L bounded by $\sup_{x,\boldsymbol{W}} G_{\boldsymbol{W}}^{(1)}(x) + \sup_{y,\boldsymbol{W}} G_{\boldsymbol{W}}^{(2)}(y) \leq \sqrt{\mathcal{J}_1} + \sqrt{\mathcal{J}_2}$. Thus, we have $\mathcal{R}_m(\mathcal{E}) \leq (\sqrt{\mathcal{J}_1} + \sqrt{\mathcal{J}_2})\mathcal{R}_m(\mathcal{E}_0 + (-\mathcal{E}_0)) \leq 2(\sqrt{\mathcal{J}_1} + \sqrt{\mathcal{J}_2})\mathcal{R}_m(\mathcal{E}_0)$. Next, we note that $\mathcal{R}_m(\mathcal{E}_0) = \mathcal{R}_m(\mathcal{H}_1 + (-\mathcal{H}_2)) = \mathcal{R}_m(\mathcal{H}_1) + \mathcal{R}_m(\mathcal{H}_2)$. Using same as technique as in Lemma 4, we have $\mathcal{R}_m(\mathcal{H}_1) \leq 2D^{(1)}||\boldsymbol{w}^{(1)}||_\infty\mathcal{R}_m(\mathcal{F}_1)$ and $\mathcal{R}_m(\mathcal{H}_2) \leq 2D^{(2)}||\boldsymbol{w}^{(2)}||_\infty\mathcal{R}_m(\mathcal{F}_2)$.   $\square$

**Theorem 4** For the energy function $E(h,\boldsymbol{x},\boldsymbol{y}) = \frac{1}{2}||G_{\boldsymbol{W}}^{(1)}(\boldsymbol{x}) - G_{\boldsymbol{W}}^{(2)}(\boldsymbol{y})||_2^2$, over the input set $\mathcal{X} \in \mathbb{R}^N$, hypothesis class $\mathcal{H} = \{G_{\boldsymbol{W}}^{(1)}(\boldsymbol{x}) = \sum_{i=1}^{D^{(1)}} w_i^{(1)}\phi_i^{(1)}(\boldsymbol{x}) = \boldsymbol{w}^{(1)^T}\Phi^{(1)}(\boldsymbol{x}), G_{\boldsymbol{W}}^{(2)}(\boldsymbol{y}) = \sum_{i=1}^{D^{(2)}} w_i^{(2)}\phi_i^{(2)}(\boldsymbol{y}) = \boldsymbol{w}^{(2)^T}\Phi^{(2)}(\boldsymbol{y}) \mid \Phi^{(1)} \in \mathcal{F}_1, \Phi^{(2)} \in \mathcal{F}_2, \forall \boldsymbol{x}\ ||\Phi^{(1)}(\boldsymbol{x})||_2 \leq A^{(1)}, \forall \boldsymbol{y}\ ||\Phi^{(2)}(\boldsymbol{y})||_2 \leq A^{(2)}\}$, and output set $\mathcal{Y} \subset \mathbb{R}^N$, if the feature set $\{\phi_1^{(1)}(\cdot), \cdots, \phi_{D^{(1)}}^{(1)}(\cdot)\}$ is $\vartheta^{(1)}$-diverse with a probability $\tau_1$ and the feature set $\{\phi_1^{(2)}(\cdot), \cdots, \phi_{D^{(2)}}^{(2)}(\cdot)\}$ is $\vartheta^{(2)}$-diverse with a probability $\tau_2$, then with a probability of at least $(1 - \delta)\tau_1\tau_2$, the following holds for all h in $\mathcal{H}$

$$
\begin{aligned}
\mathbb{E}_{(\boldsymbol{x},\boldsymbol{y})\sim\boldsymbol{D}}[E(h,\boldsymbol{x},\boldsymbol{y})] \leq & \frac{1}{m} \sum_{(\boldsymbol{x},\boldsymbol{y})\in\boldsymbol{S}} E(h,\boldsymbol{x},\boldsymbol{y}) \\
& + 8(\sqrt{\mathcal{J}_1} + \sqrt{\mathcal{J}_2})\left(D^{(1)}||\boldsymbol{w}^{(1)}||_\infty\mathcal{R}_m(\mathcal{F}_1) + D^{(2)}||\boldsymbol{w}^{(2)}||_\infty\mathcal{R}_m(\mathcal{F}_2)\right) \\
& + \left(\mathcal{J}_1 + \mathcal{J}_2\right)\sqrt{\frac{\log(2/\delta)}{2m}},
\end{aligned} \tag{35}
$$

where $\mathcal{J}_1 = ||\boldsymbol{w}^{(1)}||_\infty^2 \left(D^{(1)}A^{(1)^2} - \vartheta^{(1)^2}\right)$ and $\mathcal{J}_2 = ||\boldsymbol{w}^{(2)}||_\infty^2 \left(D^{(2)}A^{(2)^2} - \vartheta^{(2)^2}\right)$.

*Proof.* We replace the variables in Lemma 1 using Lemma 10 and Lemma 11.   $\square$

## J   Image Generation Example Settings and Additional Results

For the EBM model, we used a simple CNN model composed of four convolutional layers followed by a linear layer. The full CNN model is presented in Table 3. The training protocol is the same as in (UvA; Du & Mordatch, 2019), i.e., using Langevin dynamics MCMC and a sampling buffer to accelerate training. All models were trained for 60 epochs using Adam optimizer with learning rate $lr = 1e - 4$ and a batch size of 128. In addition to the results presented in the paper, Figure 5 presents additional qualitative results. For the first two examples (top ones), the model is able to converge to a realistic image within reasonable amount of iterations. For the last two examples (in the bottom), we present failure cases of our approach. For these two tests, the generated image still improves over iterations. However, the model failed to converge to a clear realistic MNIST image after 256 steps.

| Layer | Output shape |
|---|---|
| Input | [1,28,28] |
| Cov (16 5 × 5) | [16,16,16] |
| Swish activation | [16,16,16] |
| Cov (32 3 × 3) | [32,8,8] |
| Swish activation | [32,8,8] |
| Cov (64 3 × 3) | [64,4,4] |
| Swish activation | [64,4,4] |
| Cov (64 3 × 3) | [64,2,2] |
| Swish activation | [64,2,2] |
| Flatten | [256] |
| Linear | [64] |
| Swish activation* | [64] |
| Linear | [1] |

Table 3: Simple CNN model used in the example. * refers to the features' layer.

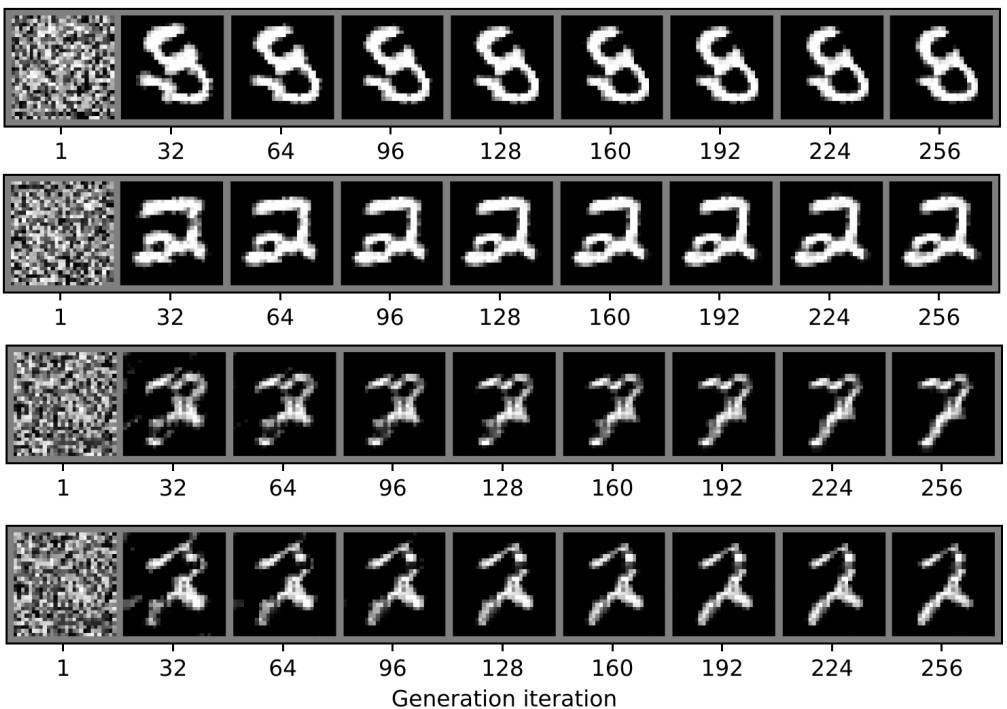

Figure 5: Qualitative results of EBM augmented with our regularizer with $\beta == 1e^{-13}$: Few intermediate samples of the MCMC sampling (Langevin Dynamics).

