# OpenReview forum: "On Feature Diversity in Energy-based Models"
_ICLR.cc/2023/Conference — Submitted to ICLR 2023_

### Official Review · Reviewer_mRZM · 2022-10-24

**Confidence:** 4
**Clarity, Quality, Novelty And Reproducibility:** See the comments above
**Correctness:** 3
**Technical Novelty And Significance:** 2
**Empirical Novelty And Significance:** 2
**Recommendation:** 5

**Strength And Weaknesses:**

### Strength
- This paper studies the generalization ability of EBMs through the lens of feature diversity, which is important due to the wild applications of EBMs
- The proposed theory and method are clearly motivated and described.
- The theoretical proof is based on Rademacher complexity and is easy to follow.
- It successfully deduces the upper-bounds for three different forms of EBMs under the unifying framework and the experimental results are promising.
### Weaknesses
- The deduced upper-bounds are typically loose and might not be able to guide the practice. Hence, it is better to have some theoretical or intuitive explanations about how tight those upper-bounds are.
- The experiments in this paper is kind of insufficient. This paper only includes some experimental results on image generation and continual learning, which are not consistent with its theoretical proof of regression, classification, and implicit regression tasks. Therefore, experimental results on the above three tasks should be reported clearly to verify the theory. Besides, the authors should conduct some experiments on more realistic and challenging datasets, such as ImageNet, CelebA etc.

- Comparison between prior work is lacked . For example, EBGAN [Zhao et al. 2017] uses a similar regularization based on cosine distance for diversity generation.

- Validity of the diversity definition 1.   I am wondering whether Def. 1 is a good measure of diversity, especially when the diversity should be used for a set of samples $S=\{ x_1, … , x_n \}$, as indicated in Eq. 16.  In Eq. 16, you simply sum of the contributions of all samples. However, some holistic information might be lost in this way.

There are well-established diversity tools in the literature, for example, the DPPs, see a nice survey and the references therein [Kulesza- Taskar 2012]. To apply DPP here,  for one feature $i$,  it is sensible to take the response of all n samples as its representation vector  $[\phi_i(x_1), … \phi_i(x_n)]^T$, then you can construct the DPP diversity metric by choose some kernel measuring the similarity. This might give rise a more reasonable diversity metric since DPP has a clear interpretation as the square of  volume spanned by the representation vectors.

Can you comment on the above possibilities?

- There seems to be some misuse of symbols in the paper. For example, it uses the sample symbol R_m to represent both empirical Rademacher complexity (in Definition 2) and the standard Rademacher complexity (in Lemma 2). The writting needs to be further checked and polished.

### References

Junbo Zhao, Michael Mathieu, and Yann LeCun. Energy-based generative adversarial network. ICLR, 2017.

Alex Kulesza, Ben Taskar, et al. Determinantal point processes for machine learning. Foundations and Trends® in Machine Learning, 5(2–3):123–286, 2012



**Summary Of The Paper:**

This paper studies the influence of feature diversity on the generalization of energy-based models (EBM), which refers to the gap between the estimated energy function and the true energy distribution.  The authors propose to improve the performance of Energy-Based Models (EBMs) by enhancing the feature diversity of the inner model. It provides a PAC theory upper-bound (correlated with the feature diversity) of the gap between the true and empirical expectation of the energy for three kinds of tasks. To verify the effectiveness of the upper-bounds, it designs a simple regularization algorithm, which shows performance gain compared with standard EBMs.

**Summary Of The Review:**

This paper proposes an interesting theory analysis and improvement for EBMs by strengthening the feature diversity. However, considering the upper-bounds are loose and there are insufficient experimental results, the paper needs to be further improved.

---

### Official Review · Reviewer_AdvF · 2022-10-24

**Confidence:** 2
**Correctness:** 3
**Technical Novelty And Significance:** 2
**Empirical Novelty And Significance:** 2
**Recommendation:** 5

**Clarity, Quality, Novelty And Reproducibility:**

The paper is easy to follow, while the experiemental evaluations are weak and insufficient.

**Strength And Weaknesses:**

Strength

- Well-organized paper with strong motivation.
- The proof framework is unifying, explicit, and easy to follow.
- The proposed simple regularization method is easy to be implemented and validated, and the experimental results are promising.

Weaknesses
- The empirical evidence is not strong enough to show a direct correlation between the generalization error of energy-based approaches and the v-diversity defined in the paper. On one hand, the proposed regularization algorithm is only tested on some simple image datasets, e.g. MNIST, and CIFAR. On the other hand, this paper only reports limited experimental results on image generation and continual learning, which are inconsistent with the theory part, including regression, classification, and implicit regression. Therefore, in order to prove the consistency of theory and practice, authors should either provide some PAC theory proof of image generation and continual learning or conduct experiments on the regression, classification, and implicit regression tasks and report the detailed results compared with standard energy-based approaches.
- The definition of v-diversity is not well explained. I do not think it can represent the diversity of the feature set since $\phi_i(x)$ and $\phi_j (x)$ are at different positions of the feature set.  Besides, in the paper, all the upper-bounds are correlated with $\sqrt{(DA^2-v^2 )}$ , and the author claims that we can deduce the upper-bounds by increasing $v$. However, as $v$ increases, $A$ also has an upward tendency since increasing $v$ could cause the increase in the magnitude of $\phi_i(x)$ . Therefore, I don 't think these upper-bounds can explain the role of v-diversity.


**Summary Of The Paper:**

This paper proposes a unifying PAC theory proof, showing the potential of improving the generalization error of energy-based approaches by reducing the redundancy of the feature set. It combines the Rademacher complexity and the definition of v-diversity to compute the upper-bound of the gap between the true and empirical expectation of the energy. Some experiments are conducted to verify its claims.

**Summary Of The Review:**

The theory proposed in this paper lacks many important explanations, including definition and conclusion parts. Furthermore, the experimental tasks are inconsistent with the theory.

---

> ### Author Response · Authors · 2022-11-18
> **Response to Reviewer AdvF**
>
> We thank the Reviewer finding our paper well-organized paper with strong motivation. We hope to provide clarifications to address the highlighted issues to raise the score of the reviewer.
>
> **Q**: The empirical evidence is not strong enough to show a direct correlation between the generalization error of energy-based approaches and the v-diversity defined in the paper. On one hand, the proposed regularization algorithm is only tested on some simple image datasets, e.g. MNIST, and CIFAR. \
> **A**: As mentioned in our response to reviewer 1, experiments conducted in this paper, serve only as proof-of-concept to show how our theory can be used in practice. The focus of the paper was mainly on theory, with a small experimental part in the end to validate our results. Moreover, note that CIFAR10/CIFAR100, in continual learning context, are the most commonly used datasets for evaluating new approaches.
>
> **Q**: On the other hand, this paper only reports limited experimental results on image generation and continual learning, which are inconsistent with the theory part, including regression, classification, and implicit regression. Therefore, in order to prove the consistency of theory and practice, authors should either provide some PAC theory proof of image generation and continual learning or conduct experiments on the regression, classification, and implicit regression tasks and report the detailed results compared with standard energy-based approaches. \
> **A**: We disagree with the reviewer about the inconsistency between the theory and the practice. The three considered cases in the theory, i.e., Figure 1, do not cover standard classification and regression only. They encapsulate several supervised and unsupervised approaches, e.g., image generation. The first task in our experiments is a regression problem. The image generation task can be interpreted as a classification problem in the EBM framework (Figure 1-a) with a fixed label y. That is, the objective is to learn an energy-model such that when X is drawn the desired distribution the output energy is low, and when it is not the output energy is high. This is exactly what is done in our image generation experiments: we train a model which takes an image X as input and outputs low energy when X comes from that distribution.  Continual learning is a classification task too (Figure 1-a), where the model takes two inputs X and y (the class label in this case). (X,y) can be any data point from the series of tasks. The model outputs a low energy if X belongs to class y.
>
> **Q**: The definition of v-diversity is not well explained...Therefore, I don't think these upper-bounds can explain the role of v-diversity. \
> **A**: There seems to be a misunderstanding of diversity as defined in our paper. A feature set $\{\phi_1(.),\phi_2(.),\cdots,\phi_N(.)\}$ in the context of our analysis refers to the set of functions are used to extract features. So, we want to model how different and diverse these features are. In other words, how the pattern extracted by $\phi_i(.)$ is different than the one extracted by $\phi_j(.)$, for $j \neq i$. Intuitively, if two feature maps $\phi_i(.)$ and $\phi_j(.)$ are non-redundant, they have different outputs for the same input with a high probability.  That is the information captured by our ($\vartheta-\tau$)-diversity. We derived generalization bounds based on this definition. The bounds are inversely proportional to $\vartheta$ showing that diversity helps improving generalization. Regarding other variables in the bounds, it is true that $\vartheta$ and $A$ are connected. But this is typically the case for most variables in generalization bounds, as all the model parameters are linked. However, our findings can be interpreted as follows: Given two models with the same value of $A$ (maximum $L_2$-norm of the features),  the model with higher diversity $\vartheta$ has a lower generalization bound and is likely to generalize better. To avoid confusion, we now add this analysis in the general discussion in section 2.1.

---

### Official Review · Reviewer_VPpj · 2022-10-24

**Confidence:** 4
**Correctness:** 3
**Technical Novelty And Significance:** 1
**Empirical Novelty And Significance:** 1
**Recommendation:** 1

**Clarity, Quality, Novelty And Reproducibility:**

The theory part of the current submission is almost the same as the workshop paper. The only new addition is the experiment section. So, to my understanding, the contribution is mainly empirical study, and the novelty is low.

The main algorithmic change is the regularization (Eq. 16), which is straightforward to implement.


**Details Of Ethics Concerns:**

The theory part of the current submission is almost the same as the workshop paper. There is no citation to the workshop paper or mentioning it in the paper.

Even the workshop paper can be claimed as an un-published work, the authors should at least cite it considering even Hinton’s RMSProp slides have been cited over 800 times.


**Strength And Weaknesses:**

Strength:

1. The paper is well organized and well written.
2. The extension of the PAC theory of EBMs of Zhang et al. to feature diversity analysis is interesting and looks solid. But frankly, it’s hard for me to fully verify all the mathematical derivations given such a short ICLR review timeline.

Weakness:

1. I have been working in this area for a while. It happened to me that I read a similar publication a year ago that has the same title (https://openreview.net/forum?id=ks3Q08yy66rv). The theory part of the current submission is almost the exact copy & paste of the workshop paper above. The only new addition is the experiment section. So, to my understanding, the contribution of this paper is mainly empirical study.

2. The experimental evaluations are very limited to small benchmarks and non-competitive baselines. For example, the image generation is only limited to MNIST without considering at least Cifar10/100/CelebA etc. In terms of the baseline algorithms, only the EBM algorithm of Du et al 2019 is compared and all latest SOTA EBM methods are ignored.

3. Although the authors show that the gains are consistent. But they seem very small. Even on the weak baseline of CL, where EBM-CL’s accuracies on CIFAR10/100 are about 30%-40%, the improvements are insignificant.


**Summary Of The Paper:**

The paper extends the PAC theory of EBMs to analyze the impact of feature diversity on the performance of EBMs. The generalization bounds for regression, binary classification and implicit regression w.r.t. the feature set redundancy are derived. Experimental results on MNIST for image generation and Cifar10/100 for continual learning are provided.

**Summary Of The Review:**

I’d recommend a strong rejection of the paper. The reasons are two-fold:
1. The theory part is almost exact the same to the workshop paper of 2021.
2. The empirical evaluations are significantly weak.

Even the workshop paper can be claimed as an un-published work, the authors should at least cite it considering even Hinton’s RMSProp slides have been cited over 800 times.

---

### Official Review · Reviewer_oTpp · 2022-10-25

**Confidence:** 3
**Correctness:** 2
**Technical Novelty And Significance:** 3
**Empirical Novelty And Significance:** 3
**Recommendation:** 5

**Clarity, Quality, Novelty And Reproducibility:**

**Clarity**

The paper is clear and well-written. I enjoyed the reading.

**Quality**

The methodology used in the theoretical analysis is correct and the results are sound. However, some of the claims are not well supported.
The evaluation methodology is also technically correct.

**Novelty**

The paper proposes a new regulariser to promote feature diversity in energy-based models and an accompanying theory motivating its need.
The theory sheds new lights on the generalisation performance of energy-based models and it opens up to new directions for regularising such models. Additionally, it can potentially connect to recent work using the principle of redundancy minimisation in self-supervised learning.

**Reproducibility**

Code is not available.

**Strength And Weaknesses:**

**Strenghts**
- Regularization in energy-based models is an important and relevant problem.
- The proposed regulariser is to my knowledge new and can potentially link to recent work on redundancy reduction criteria used in self-supervised learning.
- The theory for the regulariser is simple and at the same time elegant and it provides a solid motivation on the need of the regulariser. Additionally, I think that the analysis can trigger new discussion in the community of energy-based models and inspire new regularising approaches.
- The paper is well-written and clear. I enjoyed reading it.

**Weaknesses**
- The claim that “the theory developed in our paper is agnostic to the loss function” is not correct (this appears in several parts of the paper). Indeed, note that the contributing term on the Rademacher complexity of the energy function in Theorem 2 (regression, using $L_1$ norm energy score) and in Theorem 3 (binary classification, using a cross-entropy-like energy score) doesn’t depend on the diversity parameter from the regulariser. Consequently, minimising the proposed regulariser doesn’t guarantee a reduction between true and empirical estimate of the average energy score for those two cases. On the contrary, it seems that the generalisation performance depend on the definition of the energy score function. Can you be more precise and elaborate on this aspect?
- The claim that “we provide theoretical guarantees for WORKS ON SELF-SUPERVISED LEARNING showing that reducing redundancy in the feature space can indeed improve the generalisation of the EBMs” is not precise. Indeed, note that the link between feature diversity and the objectives considered by negative-free approaches in self-supervised learning is not clear yet and it should be made more explicit. The latter approaches typically attempt to increase the correlation between representations of different views of the same data, while reducing their redundancy. It has been shown that this can be related to the principle of information bottleneck. Would it possible to make a clear link between the proposed diversity regulariser and the information bottleneck? This would strenghten the value of the proposed theory.
- Code is not available


**Minor questions**
- In proof of Lemma 4, shouldn’t you consider using the $L_2$ norm in order to be consistent with the definition of the energy function?
- The range of values for the hyper parameter beta in the regulariser is weird. Do you have an intuition on why you consider such small values? Should this range be considered also in other tasks)?


**Summary Of The Paper:**

The paper provides a new regulariser term for training energy-based models, which promotes feature diversity.
A theoretical analysis on the generalisation performance of energy-based models using the PAC-learning framework gives a solid motivating evidence on the need of the regularizer.
Specifically, the authors consider an existing generalisation bound for energy-models (where the difference between true and empirical averaged energy score is upper bounded by two terms, namely the Rademacher complexity of the energy function and its supremum value) and extend the theory by expressing the two terms in the bound as a function of the parameter involved in the new regulariser term.
The analysis is carried on three different kinds of energy functions for the purposes of regression, binary classification and implicit regression.

Experiments are performed:
- On a synthetic dataset showing the benefits of including the regulariser over unregularised approaches.
- On a MNIST task for implicit density estimation, thus evaluating generative and log-likelihood performance.
- On a continual learning task on CIFAR10 and CIFAR100, thus providing evidence on the improved predictive performance.


**Summary Of The Review:**

Overall, the paper provides a significant contribution to the literature of energy-based models. However, some of the claims are not well supported and should be made more precise. I initially recommend for a score of 6.

---- POST-REBUTTAL ----

While I think that the theory represents an interesting contribution of the work, the main concern on the bounds still remains. Indeed, the diversity parameter only controls one addend in the bounds of Eq. (13) and Eq. (14). Its maximization doesn't necessarily reduce the difference between expected and empirical energy scores. Additionally, as pointed out by reviewer AdvF, increasing the diversity paramater might indirectly increase the magnitude of the feature set, consequently increasing $A$ in the bounds and "cancelling its positive effect".
In my opinion, strengthening these aspects would suffice to accept the paper (even without an extensive experimental analysis). However, since the theory needs additional explanations/modifications, I consider the paper incomplete and not ready for publication yet. Therefore I change my score to 5.

---

> ### Author Response · Authors · 2022-11-18
> **Response to Reviewer oTpp**
>
> We thank the Reviewer for finding our paper clear and well-written.
>
> **Q**: The claim that “the theory developed in our paper is agnostic to the loss function” is not correct...Can you be more precise and elaborate on this aspect? \
> **A**:  There seems to be a misunderstanding of the theory part. Theorems 1-4 bound the generalization gap of the energy for EBM. First note that the Rademacher complexity appearing in Theorems 1-4 is not the Rademacher complexity of the feature space and not of the full model. The bounds found are inversely proportional to $\vartheta$. Showing that the  higher the diversity ($\vartheta$), the lower the upper bound is. This shows that promoting diversity can improve generalization. Note that the energy function and the loss refer to two different entities in the context of ERM. The theory conducted and the proofs are independent of the loss used and of the optimization technique: given the energy-function, the analysis is valid for any loss and any optimization method used. Thus the claim “the theory developed in our paper is agnostic to the loss function and the optimization". For the proposed regularizer, we merely used a simple approach, based on a relaxed-version of Definition 1, showing one way that can be followed for using our theory in practice, which confirms that promoting diversity indeed can help improve performance.
>
> **Q**:  The claim that “we provide theoretical guarantees for WORKS ON SELF-SUPERVISED LEARNING showing that reducing redundancy in the feature space can indeed improve the generalization of the EBMs” is not precise... This would strengthen the value of the proposed theory.\
> **A**: Thanks for raising this interesting question.  Our intuition is that self-supervised learning using redundancy reduction, e.g., Barlow-Twins, can be seen within the EBM framework as an implicit regression task where the model takes two input images X and Y, where Y is an augmented version of X. The energy function in this case is modeled with the redundancy between the individual components of the embedding vectors.  Note that our definition of diversity (non-redundancy) is slightly different than the one used in Barlow-Twins. In this paper, we have introduced a theoretical definition of diversity and presented theoretical tools on how to study generalization under diversity. We showed that reducing redundancy in the feature space can indeed improve the generalization of EBM.  It is our belief that this is the first step toward theoretically studying how Barlow-Twins helps generalization using PAC-theory. In our work, we merely made the connection between our theory and self-supervised learning using redundancy reduction. To avoid confusion, we relaxed our claim in the revised manuscript as follows: `we provide theoretical guarantees that reducing redundancy in the feature space can indeed improve the generalization of the EBM. This can pave the way toward providing theoretical guarantees for works on self-supervised methods'. Connection to the information bottleneck is also interesting but outside the scope of this work.
>
>  **Q**: Code is not available \
> **A**: Now the codes are added to the supplementary material.
>
> **Q**: In proof of Lemma 4, shouldn't you consider using the $L_2$ norm in order to be consistent with the definition of the energy function? \
> **A**:  As specified in first paragraph of section 2.1, for the regression task, i.e., Lemmas 3-7 and Theorems 1-2, we considered the 1-D case. i.e., $y,h(x) \in  \mathbb{R} $. So, in this case,  the norm 2 $||.||_2 $ and $|.|$ are equivalent. Note it is straightforward to extend the findings to $ \mathbb{R}^D$ by using $||.||_2 $ in the proofs.
>
> **Q**: The range of values for the hyper parameter beta in the regularizer it weird. Do you have an intuition, why you consider so small values? And should this range be considered in practice (in other tasks)? \
> **A**:  We used directly the definition of diversity as a regularizer (Eq. 16). It contains two sums: the first sum is over the whole batch and the second sum is over all pairs of units within the layers. This yields a total of $ND^2$ terms, where N is the batch size and D is the number of units within the layer. This results in empirically large values of the second term ($\sim 10^9$). Thus $\beta$ needs to be small so the loss is not dominated by the second term. Empirically, we found that $[10^{-11},10^{-13}]$ corresponds to a stable range. Note that the proposed regularizer is merely a proof of concept to show that encouraging feature diversity can improve the performance of EBMs. If we normalized with $ND^2$, the range of values of $\beta$ will be higher without any change in the results. However, when we defined our regularizer, we tried to stay as close as possible to the ($\vartheta-\tau$)-diversity definition (Definition 1) and the derived theorems, which does not include any normalization term.

---

### Official Review · Reviewer_FAvo · 2022-10-30

**Confidence:** 3
**Correctness:** 2
**Technical Novelty And Significance:** 2
**Empirical Novelty And Significance:** 2
**Recommendation:** 5

**Clarity, Quality, Novelty And Reproducibility:**

Clarity and Quality: The materials in section 1&2 need to be re-organized.
Novelty: Moderately.
Reproducibility: Should be easy to reproduce.

**Strength And Weaknesses:**

This paper has provided a way to identify the quantification of feature diversity by the proposed measurement method named “θ-diversity”. The authors extend this idea in PAC learning and show that reducing the redundancy of the feature set can improve the generalization of EBMs. Please refer to the following questions about my concerns.

Weakness:
1. For the definition 1, which is the major contribution of this paper, the definition also relies on a probability value τ. Therefore, the method should at least be called ​​θ- τ (like epsilon-delta) to avoid confusion.
2. How to define high probability in definition 1? Is it related to the distribution of x. Should this be considered in the defined boundary?
What will happen if there are only a few samples that contain similar contents, where the boundary will be close to zero in most feature functions? How do we treat the effect of the distribution of data (x) on the boundary value?
3. In formula 16, the regularization term is not just about the boundary but an integration over the entire dataset and feature set. Even the experiment could sufficiently show improvement in performance, I can’t see its relationship with the proposed diversity measurement function.
4. In the image generation experiment (3.2), the authors only apply the proposed method on a simple dataset MINIST. There are lots of works based on EBM for image/text generation on a large dataset. The author should at least show some evaluation results to prove the effectiveness of the proposed method.
5. In table 1, it looks like the selection of beta could affect the performance. The author should also conduct an experiment on the effectiveness of the proposed method in terms of beta.
6. In 3.3, what are boundary settings?

Writing:

It would be better for the authors to reorganize the paper structure in section 1&2. The introduction of energy models applied in classification, regression, or implicit regression tasks should be put in the second part as related materials while a brief introduction about PAC learning should be put in the 1st section since it has a close relationship with the contribution.

Missing important references:

The description of the advance of EBMs in the first paragraph of the paper is incomplete and lacks important pioneering works. The authors randomly cite EBM works without mentioning those important ones.  For example, the first paper that proposes to train a generative EBMs parameterized by a modern deep neural network and learned it by Langevin based MLE is in (Xie. ICML 2016) [1]. The first shallow EBM using Langevin for data generation can date back to 1998 in [2]. After the era of deep learning, the paradigm in [1] has been applied to videos [3], 3D voxels [4], point clouds [5] and scenes [6]. Supervised approaches might include supervised conditional learning [7], saliency prediction [8], and trajectory prediction [9]. New generative EBM frameworks also include coarse-to-fine EBM [12], CoopNet [10], CoopFlow [11], VAEBM [13]. Without knowing the history and the advance of EBMs, the contribution of the paper might be questionable.


[1] A Theory of Generative ConvNet. ICML 2016.

[2] Grade: Gibbs reaction and diffusion equations. ICCV 1998.

[3] Synthesizing Dynamic Pattern by Spatial-Temporal Generative ConvNet. CVPR 2017.

[4] Learning Descriptor Networks for 3D Shape Synthesis and Analysis. CVPR 2018.

[5] Generative PointNet: Deep Energy-Based Learning on Unordered Point Sets for 3D Generation, Reconstruction and Classification. CVPR 2021.

[6] Patchwise Generative ConvNet: Training Energy-Based Models from a Single Natural Image for Internal Learning. CVPR 2021.

[7] Cooperative Training of Fast Thinking Initializer and Slow Thinking Solver for Conditional Learning. TPAMI 2021.

[8] Energy-Based Generative Cooperative Saliency Prediction. AAAI 2022

[9] Energy-Based Continuous Inverse Optimal Control. TNNLS 2022

[10] Cooperative Training of Descriptor and Generator Networks. PAMI 2018

[11] A Tale of Two Flows: Cooperative Learning of Langevin Flow and Normalizing Flow Toward Energy-Based Model. ICLR 2022.

[12] Learning Energy-Based Generative Models via Coarse-to-Fine Expanding and Sampling. ICLR 2021.

[13] VAEBM: A Symbiosis between Variational Autoencoders and Energy-based Models. ICLR 2021.

**Summary Of The Paper:**

This paper has provided an analysis method to evaluate the feature representation of energy-based models based on feature diversity. The authors extend the probably approximately correct (PAC) theory in the view of redundancy reduction on the performance of energy-based models.

**Summary Of The Review:**

This paper has proposed a metric to define the diversity in feature representation and extend it in the loss function to improve the model generalization. However, the author hasn’t given enough proof and reason to support the effectiveness of the proposed method. Furthermore, as shown in formula (16), this could be used as a general term in supervision of deep learning models, but not necessarily to energy-based models. Besides, the experiments are only conducted on some simple dataset and are not sufficient to demonstrate the universe effectiveness of the proposed method.

--Post Rebuttal
I change my score from 3 -> 5 due to the explanation and revision from the author. Although the author has claimed that the experiment part is only to show how the proposed regularization term affects generalization in the EBM context, I still feel it necessary to see experiments on another task/dataset except for the CL (as proposed by reviewer mRZM). Moreover, if the author could also answer the concerns pointed out by reviewer AdvF and oTpp where the proposed regularization term will eventually reduce $\sqrt{\left(D A^2-v^2\right)}$, I'd like to further move my ratings.

---

> ### Author Response · Authors · 2022-11-18
> **Response to Reviewer FAvo**
>
> We hope to provide clarifications to address the highlighted issues. Please, let us know if there are further questions.
>
> **Q**: 1- For the definition 1, the definition also relies on a probability value $\tau$. Therefore, the method should at least be called $\vartheta-\tau$ (like epsilon-delta) to avoid confusion. \
> **A**: Thanks for the suggestion. We agree that it makes more sense to call it ($\vartheta-\tau$) instead of $\vartheta$. It was renamed in the revisited manuscript.
>
> **Q**:2- How to define high probability in definition 1? Is it related to the distribution of x. Should this be considered in the defined boundary? What will happen if there are only a few samples that contain similar contents, where the boundary will be close to zero in most feature functions? How do we treat the effect of the distribution of data (x) on the boundary value? \
> **A**: As explained in the paper (l58 to l62), in our theoretical analysis we consider a relaxed variant of the following assumption `H*: There exists a lower bound to the distance, valid for any input x'. H* is impractical especially if the intermediate layer has ReLU activations (there is a high likelihood that there exists a certain input such that all units within the layer have zero activations and, thus,  would be theoretically zero or very small). This is due to the fact that it is defined over the entire input space. This is why we considered a relaxed variant of this assumption by introducing the relative probability: ``With a high probability $\tau$, the sum of the pair-wise distance between the output of units, is lower bounded by $\vartheta$ for any input''. The probabilistic item in this assumption is the input. From a practical point of view, the assumption made in the paper allows $\vartheta$ to be strictly positive and high. Thus, it makes the theoretical findings useful.
>
> **Q**: 3- In formula 16, the regularization term is not just about the boundary but an integration over the entire dataset and feature set. I can’t see its relationship with the proposed diversity measurement function. \
> **A**: In the theory, $(\vartheta-\tau)$-diversity can be interpreted as the lower bound of the minimum distance between a pair of features in the features set. Note that, from a practical point of view, it is a common practice to maximize the minimum by maximizing the average or the total sum, as the latter is usually smoother. This is what is done in Eq-16, we add a regularizer that maximizes the sum of the distances between the pairs. The proposed regularizer can be seen as a relaxed form of a stricter regularizer using the minimum distance. Note that here we do not claim the proposed regularizer is the best approach to maximize $\vartheta$. We only provide a simple approach showing how our theory can be used in practice. The goal is to show that indeed promoting diversity helps. Investigating more advanced approaches to encourage diversity (maximize $\vartheta$) is a promising research direction.
>
>
> **Q**: 4- Insufficient Experimental results \
> **A**: Here, we would like to stress again that the proposed regularizer and the experiments conducted in the paper are merely a proof-of-concept of the developed theory. The main aim is to show how to theoretically model diversity and to show how it affects generalization in EBM context. In the experiments,  we complement the findings with a simple regularizer, inspired by Definition 1, which promotes diversity. We test the proposed regularizer on a toy example and image generation with MNIST. Then we tested in on a more complex example:  continual learning. Note that for this task, CIFAR10 and CIFAR100 are the most commonly used datasets for evaluating new approaches.
>
> **Q**: 5- The hyperparameter $\beta$ \
> **A**:  EBMs in general are highly sensitive to hyper-parameters, and $\beta$ is a hyper-parameter. While the improvements in image generation with MNIST are marginal as this is a dataset used for illustrating the properties of new methods in simple settings, in the experiments on the more complex example (continual learning with CIFAR10-CIFAR100) the improvement is large. The regularizer achieves better performance for all the values of $\beta$.

---

> > ### Author Response · Authors · 2022-11-18
> > **Response to Reviewer FAvo  part 2**
> >
> >
> > **Q**: 6-In 3.3, what are boundary settings?  \
> > **A**: For the continual learning example, as mentioned in our paper, we use exactly the same experimental setting as in [Li et al. 2020]. Typically, a continual learning problem is set up as a sequence of distinct tasks with clear boundaries (between the tasks) that are known to the model (the boundary-aware setting). Most existing continual learning methods rely on these known boundaries for performing certain consolidation steps (e.g., calculating parameter importance or updating a stored copy of the model). The boundary-agnostic setting is more realistic and this assumption is lifted: data distributions gradually change without a clear notion of task boundaries. While many common CL methods cannot be used without clear task boundaries, EBMs can be naturally applied to this more challenging setting. To make the paper more self-contained, we added the definition of both settings in the end of first paragraph of Section 3.3.
> >
> > **Q**: It would be better for the authors to reorganize the paper structure in section 1\&2. The introduction of energy models applied in classification, regression, or implicit regression tasks should be put in the second part as related materials while a brief introduction about PAC learning should be put in the 1st section since it has a close relationship with the contribution. \
> > **A**: We thank the reviewer for the proposal. However, the main focus of this work is energy-based models. This is why we believe it should be the starting point of the paper. Then, in related works, we cover PAC in EBM context.
> >
> > **Q**: The description of the advance of EBMs in the first paragraph of the paper is incomplete and lacks important pioneering works.\
> > **A**:  The reviewer makes a valid point regarding the related works. We revisited the introduction and incorporated the mentioned references.

---

### Decision · Program_Chairs · 2023-01-20

**Decision:**

Reject

**Justification For Why Not Higher Score:**

The paper cannot be accepted because of insufficient experiments and some math issues on the upper bounds.

**Justification For Why Not Lower Score:**

N/A

**Metareview: Summary, Strengths And Weaknesses:**

This paper studies the influence of feature diversity on the energy-based models (EBM) and proposes to improve the performance of Energy-Based Models (EBMs) by enhancing its feature diversity. Specifically, the paper provides a PAC theory upper-bound of the gap between the true and empirical expectation of the energy for three kinds of tasks. To verify the effectiveness of the upper-bounds, the authors design a simple regularization algorithm to reduce the feature redundancy, and show performance gain compared with standard EBMs. The paper receives a total of 5 reviews.  The rebuttal hasn’t addressed the major concerns, including insufficient experiment (raised by Reviewers mRZM, VPp, and FAvoj) and math issue in the upper-bound (raised by Reviewers AdvF, oTpp, FAvo),  of the reviewers and all reviewers lean to reject the paper. The AC thinks that even though the paper studies an interesting and important problem for energy-based learning and the preliminary experiments also show some performance gain from the proposed method, there remains some major concerns that lead to a rejection to the current version.  Given the promising results of the current paper, the AC urges the authors to further improve their paper by taking into account all the suggestions provided by the reviewers, and then resubmit it to the next venue.


**Summary Of Ac-Reviewer Meeting:**

The internal discussion and rebuttal show that all reviewers agree on rejecting the paper. There is not any divergence among all final recommendations.